# Seasonal Changes in the Prediction Accuracy of Hayfield Productivity Using Sentinel-2 Remote-Sensing Data in Hokkaido, Japan

Ruka Kiyama [1] and Yoshitaka Uchida [2,*]

1 Graduate School of Global Food Resources, Hokkaido University, Sapporo 060-8589, Japan
2 Research Faculty of Agriculture, Hokkaido University, Sapporo 060-8589, Japan
* Correspondence: uchiday@agr.hokudai.ac.jp

**Abstract:** In large hayfields belonging to intensive dairy systems, satellite remote-sensing data can be useful to determine the hayfield yield and quality efficiently. In this study, we compared the land survey data of hayfield yield, and its quality parameters such as crude protein and neutral detergent fiber digestibility (NDF), with the Sentinel-2 satellite image data for thirteen hayfield paddocks in Kamishihoro region, Hokkaido, Japan. Commonly used indices derived from the satellite image data, including the Normalized Difference Vegetation Index (NDVI) and Enhanced Vegetation Index (EVI), were used to assess the hayfield yield and quality. In this region, hayfields are usually harvested twice yearly, in early summer (first harvest) and late summer (second harvest). As result, the Sentinel-2 data could predict the pasture growth and quality for the first harvest better than those for the second harvest. The EVI and the index based on the bands B8a and B7 were the best predictors for the biomass and NDF for the first harvest, respectively. However, the satellite-image-based predictors were not found for the second harvest. Towards the second harvest season, the color of the hayfield surface became more heterogeneous because of the flowering of weeds and uneven pasture growth, which made it challenging to predict pasture growth based on the remote-sensing data. Our land survey approach (quadrat-based sampling from a small area) should also be improved to compare the remote-sensing data and the pasture with uneven growth.

**Keywords:** satellite images; Sentinel-2; pasture productivity; pasture heterogeneity

## 1. Introduction

The efficient management of hayfield pasture as a self-sufficient feed is critical from both sustainable and economic perspectives in the livestock industry of Japan. One of the reasons for the importance is that the livestock industry in Japan relies on imports for approximately 75% of its total digestible nutrients-based feed [1]. Thus, the establishment of livestock feed production systems less susceptible to overseas feed production and economic conditions is critically needed [2]. In dairy industries in recent years, the number of dairy cows in Hokkaido (1 of the 47 prefectures in Japan, which produces more than half of the milk in Japan) has increased, and the amount of concentrate feed used for dairy cows has also increased. Despite the increasing demands, the unstable global situation and the yen depreciation have markedly increased imported feed prices [3]. Therefore, improving pasture productivity within Hokkaido is an urgent issue [4,5].

One of the essential technologies for efficient pasture management is to accurately measure the quantity and quality of pasture at the field level [6,7]. For example, in Hokkaido, Japan, pasture quality, such as its digestible nutrient contents and crude protein contents, rapidly declines when pasture yields exceed a certain level. Thus, optimizing the harvest timing has been a challenge for dairy industries [8]. To date, the dairy farming in the Hokkaido region has prioritized pasture yields over its quality because it has been economically feasible to adjust the feed quality by mixing the harvested pasture with the

imported grains [3]. However, as noted above, techniques for estimating pasture quality are becoming more important because efforts are being made to reduce the cost of imported grains used in dairy farming [9].

Thus, it will be ideal if the quantity and quality of hayfield pasture can be predicted in near real-time before harvest [10]. The real-time prediction of the hayfield quantity and quality would allow for applications such as determining the timing of the harvest based on these predictions and improving the efficiency of reducing imported concentrate feed. Conventional methods for evaluating pasture productivity are based on ground surveys and interviews with farmers. The ground surveys for pasture yields involve cutting quadrats for pasture, weighing, and drying. In addition, laboratory-based chemical analysis is used to measure pasture quality, such as protein content [11]. While these methods are accurate, they are time- and labor-intensive and require many samplings to assess a wider area [12].

On the other hand, satellite remote sensing can assess pasture productivity with much less labor than conventional methods such as ground surveys and interviews [12–14]. In addition, satellite remote sensing can obtain almost real-time information with high frequency and provide information on differences in plant conditions that cannot be determined by the human eyes, such as near-infrared and visible light [12]. Satellite remote sensing of biomass estimation has used vegetation index-based regression models. Previous studies showed a good correlation between field measurements and vegetation indices derived from satellite data [15]. According to previous studies, the Normalized Difference Vegetation Index (NDVI) was the most used index to investigate pasture production. Still, other indices, such as Enhanced Vegetation Index (EVI), were also used [13,16,17].

However, the vegetation index-based approach is known to be region- and plant-species-specific [12]. Hayfield pastures in Hokkaido are dominated by Timothy grass (*Phleum pratensis* L.), and the optimization of the harvest timing is critical because of the rapid growth, particularly in early spring after snow melt [18]. Based on our research, few studies investigated the potential use of satellite remote-sensing data on the hayfield pasture productivity, focusing on both quantity and quality in the Hokkaido region. Thus, more studies are needed.

Sentinel-2 can be used freely and with improved resolutions when compared to other remote-sensing data. In addition, Sentinel-2 has a concise observation cycle of 5 days, and many studies have already been conducted to estimate the pasture biomass using Sentinel-2 [13,17,19]. On the other hand, the usefulness of Sentinel-2 for estimating the nutrient value of Timothy-based pastures has not yet been extensively studied, while the studies on grazed pasture species such as Perennial ryegrass (*Lolium perenne*) are more abundant [20,21]. Thus, this study aimed to analyze the relationship between vegetation indices and hayfield yield and quality to assess the productivity of pastures using Sentinel-2 satellite image data.

## 2. Materials and Methods

### 2.1. Study Plots

Thirteen hayfield pasture paddocks (A–M) for dairy in Kamishihoro Town, Hokkaido, Japan were selected for this study (Table 1). All plots were dominated by Timothy grass or Quackgrass (*Elymus repens*) and solely used for silage production (not for grazing). These fields were harvested twice a year. In paddocks A, D, E, and I, more than 10 years have passed since pasture renewal. We note that in this region, old pastures tended to be invaded by Quackgrass, dominating the pasture cover, but Timothy grass was the sown grass for all the paddocks used in this study.

**Table 1.** Overview of study plots.

| | Study Point | Area (ha) | Renewal Date | Main Pasture Species |
|---|---|---|---|---|
| A | 43.25343 N 143.26831 E | 4.62 | Before 2011 | Quackgrass |
| B | 43.25098 N 143.26704 E | 4.68 | 2018 | Timothy |
| C | 43.25214 N 143.26906 E | 2.73 | 2018 | Quackgrass |
| D | 43.25167 N 143.27086 E | 1.97 | Before 2011 | Quackgrass |
| E | 43.24998 N 143.27061 E | 1.51 | Before 2011 | Quackgrass |
| F | 43.24914 N 143.26701 E | 2.44 | 2016 | Quackgrass |
| G | 43.24855 N 143.26672 E | 2.68 | 2018 | Timothy |
| H | 43.24943 N 143.26494 E | 4.53 | 2016 | Quackgrass |
| I | 43.24989 N 143.26341 E | 4.65 | Before 2011 | Timothy |
| J | 43.24399 N 143.29668 E | 1.43 | 2011 | Timothy, Dandelion |
| K | 43.24409 N 143.2951 E | 2.39 | 2011 | Quackgrass |
| L | 43.22238 N 143.27747 E | 6.79 | 2012 | Timothy, Alfalfa |
| M | 43.22305 N 143.27351 E | 3.33 | 2011 | Quackgrass |

*2.2. Yield Survey*

The ground-level pasture growth and yield surveys were conducted on 20 April, 6 May, 18 May, 1 June, 9 June, 15 June, 8 July, 26 July, and 17 August 2021. The paddocks were harvested twice. Thus, the survey data just before the harvest were used as the data for the first and second harvests. For example, the first harvest data were taken on 9 June (4 plots) and 15 June (9 plots), and the second harvest data were collected on 17 August (13 plots). Pasture clippings to estimate the biomass yields were collected at one location per paddock using the quadrat method (0.5 × 0.5 m square). The pastures in the quadrat were clipped with a lawn clipper at the height of ground level. We only took one quadrat sample per paddock for a few reasons. One was that we intended to mimic the grass assessment method traditionally applied by the local extension officers. Another was that the pasture is usually very uniform (especially from spring to summer), and we found mainly one grass species in each paddock. A third was that weeds (such as Docks (*Rumex obtusifolius*)) appeared as large patches in specific sections within a paddock. Thus, we believe it would be challenging to accurately quantify the paddock pasture yield even by increasing the quadrat samples per paddock, especially in the autumn season when the weeds start actively growing. Then, the collected pastures were weighed and air-dried (70 °C, 48 h) to produce a dry matter yield (kg DM/hectare (ha)). The pasture height was the average of five measurements measured near the quadrat. The composition of pasture nutrients was analyzed by the Agricultural Chemical Laboratory of the Tokachi Agricultural Cooperative Union for pasture samples taken on 18 May, 1 June, 9 June, 15 June, 26 July, and 17 August. The quality of the pasture was evaluated based on data on crude protein (CP) and neutral detergent fiber (NDF) digestibility obtained by the requested analysis. The CP was measured by the Kjeldahl method and converted to CP by multiplying the total nitrogen content by a nitrogen coefficient of 6.25. The NDF was measured by the detergent method.

*2.3. Sentinel-2 Satellite Images and Image Analysis Methods*

For the satellite image data, the Level-2A Sentinel-2 satellite images (the cloud cover ratio under 38%) observed on 27 April, 7 May, 14 May, 1 June, 6 June, 13 June, 18 July, 23 July, and 20 August 2021 were used. They were obtained from the Copernicus Open Access Hub [22]. The dates of these satellite observations corresponded to the dates of the ground base pasture yield survey.

First, the average pixel values (reflectance) within each paddock were extracted from the satellite data for each band using the "Raster Analysis Tools/Zone Statistics" in QGIS 3.10. From the extracted reflectance, the Normalized Difference Vegetation Index (NDVI) [23] and Enhanced Vegetation Index (EVI) [24–26] were used to predict the pasture quantity (yield and height). Two Normalized Difference Indices (NDIs), previously

reported as the best predictors of pasture quality, namely CP and NDF [21], were calculated with the following formula (Table 2).

**Table 2.** Vegetation indices and NDIs that can be calculated using pairs of Sentinel-2 bands.

| Index | Formula | Prediction | References |
|-------|---------|------------|------------|
| NDVI | $(\text{NIR} - \text{Red})/(\text{NIR} + \text{Red})$ | Yield | [23] |
| EVI | $2.5 \times (\text{NIR} - \text{Red})/(\text{NIR} + 6 \times \text{Red} - 7.5 \times \text{Blue} + 1)$ | Yield | [24–26] |
| NDI(B11,B5) | $(\text{B11} - \text{B5})/(\text{B11} + \text{B5})$ | CP | [21] |
| NDI(B8a,B7) | $(\text{B8a} - \text{B7})/(\text{B8a} + \text{B7})$ | NDF | [21] |

For the NDVI and EVI, raster images for each paddock were created using the QGIS 3.10 "Raster Calculator". Then, the histograms of the index values within each paddock were calculated using R version 4.0.0 (R Core Team 2020) with the packages "raster" and "rgdal".

*2.4. Statistical Analysis*

The Pearson's correlation coefficients between pasture dry matter yield, CP, NDF, and the calculated satellite-image-based indices (Table 2) were obtained using R version 4.0.0 (R Core Team 2020) using the package "ggpmisc" with the significance threshold value of $p < 0.05$. First, the correlation between dry matter yield, CP, and NDF and their corresponding indices was examined using the whole data sampled throughout the experimental period (from spring to autumn). Then, we evaluated using the data only from the first and second harvest timings to examine the correlations at the harvest timings. We wanted to assess the usage potential of the satellite images to predict the variability in the pasture yield and quality at the harvest rather than the changes in the images over their growth.

**3. Results and Discussion**

*3.1. Dry Matter Yield and Quality*

Pasture height and dry matter yield increased rapidly from mid-May to early June, towards the first harvest timing. The first harvested pasture yield ranged from 4430 to 8670 kg/ha, while the second harvested pasture yield ranged from 3420 to 6360 kg/ha, with the first harvested pasture being slightly larger than the second harvested pasture yield (Figure 1a,b).

In terms of the pasture quality, CP peaked at 25–30% in mid-May and decreased to 10–15% before the first pasture was harvested. During the second pasture regrowth period, CP remained steady and ranged from 10 to 20% from July to mid-August. There was no difference between the CP of the first and second harvest pasture. The NDF increased with the pasture growth until the first harvest but showed a slight decrease after the first harvest towards the second harvest (Figure 1c,d).

Dry matter yields of Timothy-dominated pastures in this study area were previously reported to be about 5000 kg/ha for the first harvested pasture and 3000–4000 kg/ha for the second harvested pasture [4]. Thus, in our study, the dry matter yields of the first and second pasture were higher than the reported dry matter yields. We also note that a previous report stated that the productivity of Quackgrass, the dominant weed in the researched area, was lower than that of Timothy [27,28]. Still, in our study, we observed that there was no significant difference in dry matter yield between the Timothy and Quackgrass-dominated pastures in terms of quantity.

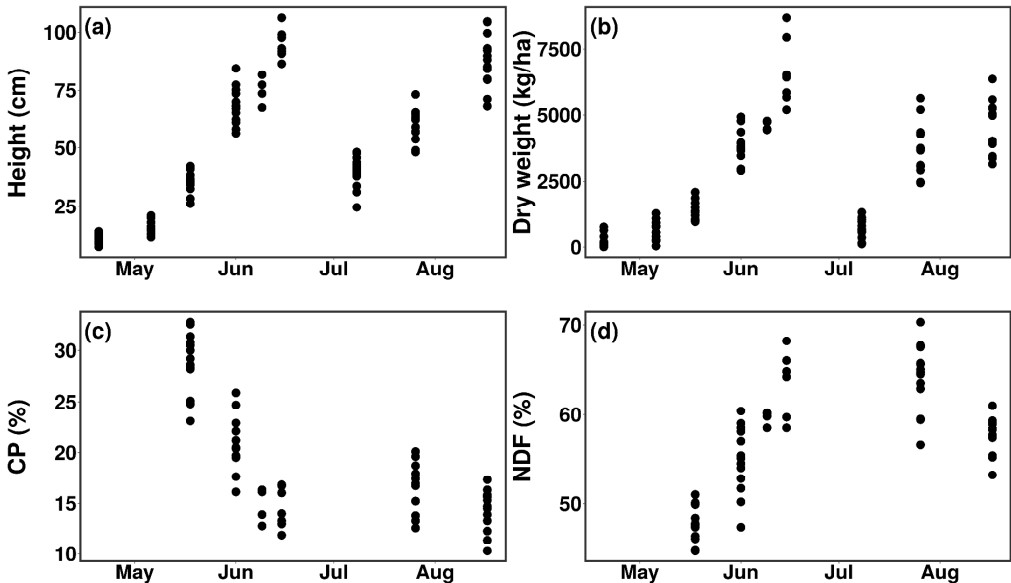

**Figure 1.** Time-series changes of (**a**): pasture height, (**b**): pasture dry matter yield, (**c**): CP, (**d**): NDF in 13 pasture plots in Kamishihoro, Hokkaido. Pasture height and dry matter yield were surveyed a total of nine times from 20 April to 17 August, and CP and NDF were analyzed six times out of all the surveys. The survey just before the first pasture was harvested was conducted on 9 June in four plots and on 15 June in seven plots. The survey just before the second pasture was harvested was conducted on 17 August in the 13 plots.

*3.2. Relationship between NDVI, EVI, and Pasture Yield over the Entire Growth Period*

Significant positive correlations ($R^2 = 0.44$ and $R^2 = 0.53$, both $p < 0.001$) were observed between the pasture dry matter yield and NDVI and EVI when the data from all survey periods were included (Figure 2). The $R^2$ values were higher for EVI than NDVI because NDVI tended to peak at an earlier stage of pasture growth, while the peak for EVI occurred at a later stage of the pasture growth period (Figure 2a,b).

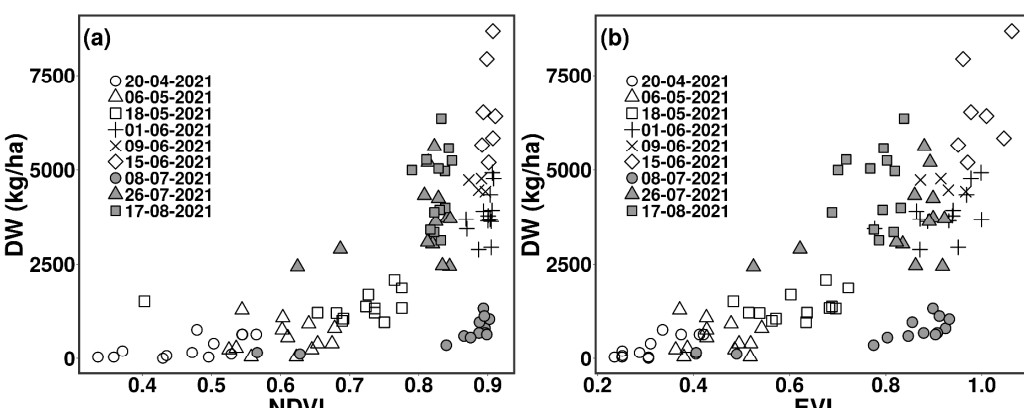

**Figure 2.** Relationships between pasture dry matter yield and NDVI and EVI. (**a**): Relationship between pasture dry matter yield and NDVI, (**b**): the relationship between pasture dry matter yield and EVI. The shape of the plot indicates the survey date, and the gray plot indicates after harvesting the first pasture.

The relationship between pasture growth and EVI or NDVI changed when the periods before the first harvest and after the first harvest were compared. During the pasture growing period before the harvest, the vegetation indices tended to increase with the increase in dry matter yield. In contrast, the same relationship was not observed after the first harvest. As the reason for this difference, we observed that during the period between

the snow melt to the first harvest, the hayfield pasture clearly changed color from brown to green (Figure 3a). Thus, the satellite images could clearly indicate the pasture growth pattern until the first harvest. On the other hand, the NDVI and EVI values were already high immediately followed by the harvest of the first pasture, even though there was very small biomass (8 July (gray circles) in Figure 2). Furthermore, we visually observed that even immediately after the first harvest, the hayfield pasture appeared green (Figure 3b) due to the stems of pasture and the pasture leaves harvested but left on the ground. Thus, we must consider the effect of pasture management and seasons to estimate the pasture biomass using satellite images accurately.

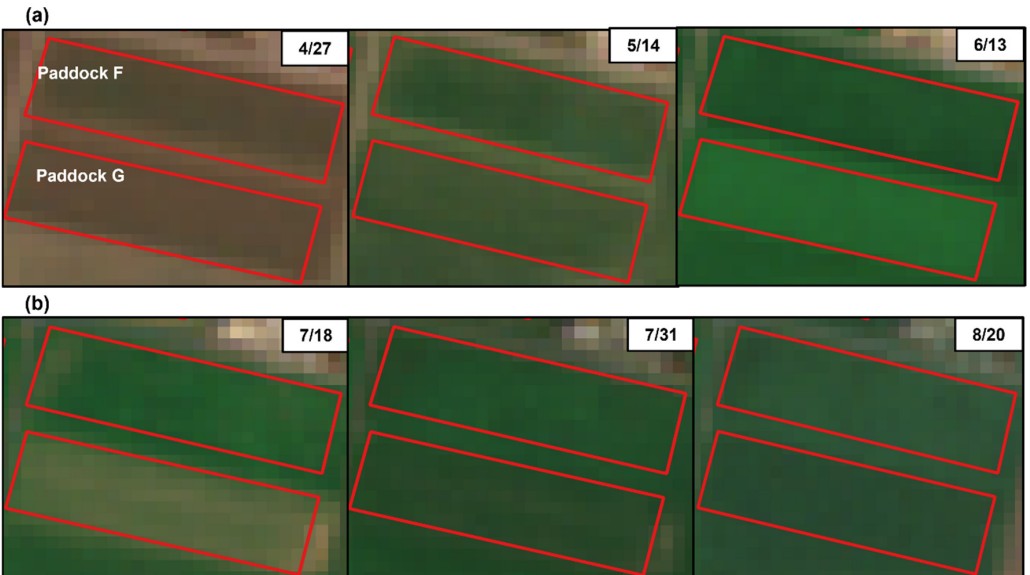

**Figure 3.** Satellite images (true color image) from Sentinel-2 of paddock F and G. (**a**): Images of the first pasture growing period from early spring to first harvesting, (**b**): images of the second pasture growing period from the early stage of regrowth after first harvesting to second harvesting.

### 3.3. Relationship between NDVI, EVI, and Pasture Dry Matter Yield at Pasture Harvest

In this section, the analysis focuses on the specific timings for the first and second harvest, in contrast to the previous section, which focused on the time course changes in pasture growth. During the first pasture harvest timing, the EVI showed a positive correlation with pasture dry matter yield. Still, during the second pasture harvest survey, there was no correlation between NDVI and EVI and pasture dry matter yield (Figure 4). At the first harvest timing, the range of observed NDVI values across the sampling plots was much narrower when compared to EVI, suggesting that the NDVI values were saturated before the harvest timing in our researched region. A previous study also noted that the EVI is a vegetation indicator less sensitive to atmospheric effects. It does not saturate values as rapidly as NDVI due to its sensitivity to high-density vegetation [29]. Therefore, EVI may provide a more accurate assessment of pasture quantity than NDVI in our study.

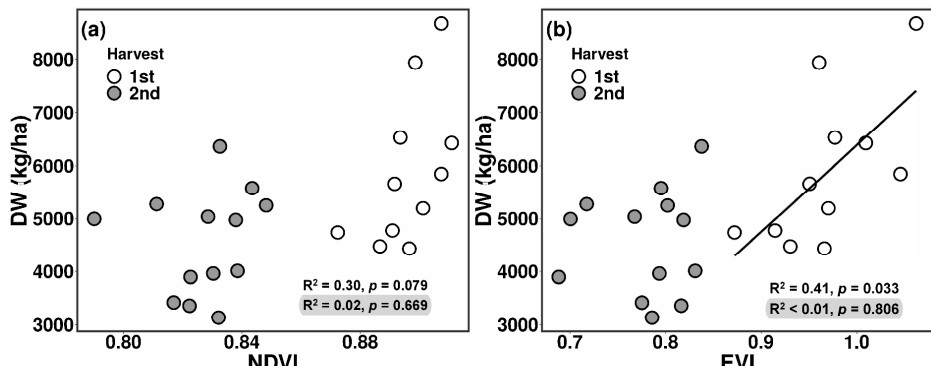

**Figure 4.** Relationships between the first and second harvested pasture dry matter yield, NDVI, and EVI at harvesting period. (**a**): Relationship between the pasture dry matter yield and NDVI, (**b**): relationship between the pasture dry matter yield and EVI. ○ indicates the first pasture harvest survey (9 and 15 June), and ● indicates the second pasture harvest survey (17 August).

*3.4. Seasonal Differences of Heterogeneity of Paddocks Based on Pixel Value Data of NDVI and EVI*

One of the reasons for the difficulty in predicting the second harvest yield based on satellite image indices such as NDVI and EVI was the large heterogeneity of the pixel value data of the NDVI and EVI within each plot. The pixel values of NDVI and EVI within a paddock tended to be more scattered, and the values tended to be lower at the second harvest than at the first harvest (Figures 5 and 6). The scattered pixel values were due to the differences in the heterogeneous regrowth after the first harvest in each paddock. Towards the end of summer, we observed increased activities of plant species other than the main sown grass (Timothy). For example, Quackgrass and Docks (*Rumex obtusifolius*) were actively growing on some of the paddocks after the first harvest. Previous studies reported that the regrowth rate of Quackgrass after the first harvest is faster than that of Timothy, especially during high summer temperatures [30]. Thus, we believe that our ground surveys (quadrat method) should have been performed with more replications to estimate the second harvest pasture. However, we note that it was markedly challenging to find the best part of a paddock where the quadrat sampling method could represent the plant biomass for the whole paddock area, especially in the autumn season because the weed species grow in large patches within some paddocks.

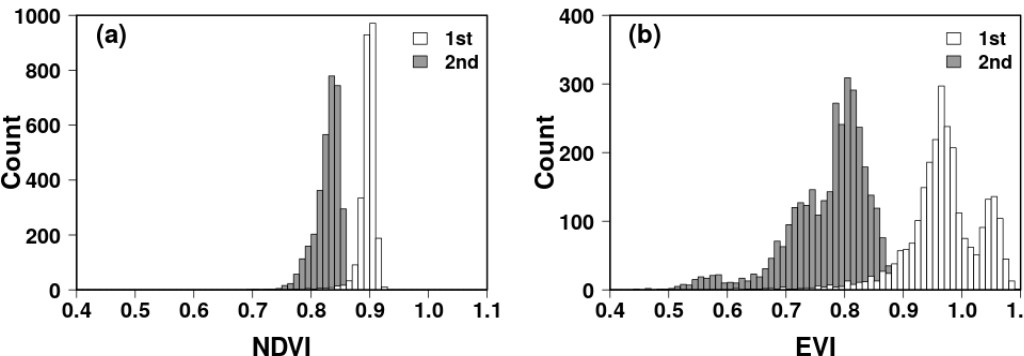

**Figure 5.** Histogram representing the pixel values of NDVI and EVI. (**a**): The distribution of the pixel values of NDVI at the harvest period, (**b**): the distribution of the pixel values of EVI at the harvest period. The white bar indicated the frequency of the pixel values of NDVI and EVI at the first harvest. The gray bar indicates the frequency of NDVI and EVI at the second harvest.

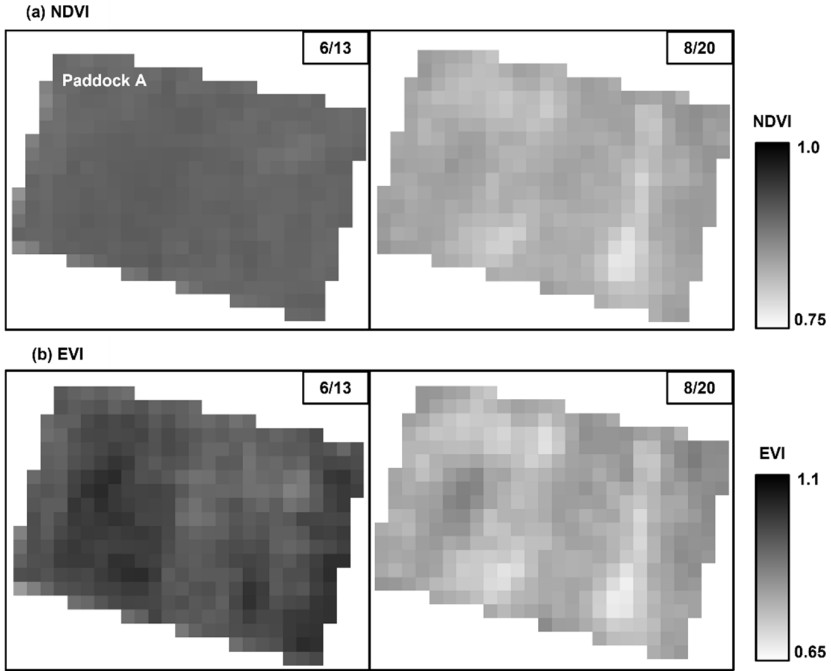

**Figure 6.** NDVI and EVI images from Sentinel-2 of paddock A on 6/13 (the first harvest) and 8/20 (the second harvest). (**a**): NDVI images with a value range from 0.75 to 1, (**b**) EVI images with a value range from 0.65 to 1.1.

*3.5. Relationship between Pasture Quality and Normalized Indices Calculated from Satellite Images*

We tested two NDIs previously reported to be the best predictors of the pasture quality data using the Sentinel-2 images ((B11 − B5)/(B11 + B5) and (B8a − B7)/(B8a + B7)) [21]. We also tested NDVI and EVI to predict CP and NDF, but there were no significant correlations between these indices and the pasture quality parameters (data not shown). Therefore, we tested the best-modeled NDIs presented by [10]. For the prediction of CP, the NDI(B11 − B5)/(B11 + B5) was the best combination.

In our study, there was no correlation between CP and NDI(B11 − B5)/(B11 + B5) values obtained from the sampling at six different timings between 18 May to 17 August (Figure 7a). We analyzed using the data only for the harvest timings, but we could not find a significant correlation between CP and NDI(B11 − B5)/(B11 + B5) (Figure 7b). The prior study reported that the B11, combined with red-edge bands, were sensitive to CP. However, our study did not show a good relationship between NDI(B11 − B5)/(B11 + B5) and CP, and it can be because of the different range of the CP values between the previous study [21] and our study. In our study, there were no samples with a CP value of under 10%, which was not the case in the previous study [21]. Thus, the pasture with lower CP values might show a better relationship with the satellite images, although this needs further testing.

Similarly, no correlation was recognized between NDF and NDI(B8a − B7)/(B8a + B7) values from all six surveys performed between 18 May and 17 August (Figure 7c). On the other hand, a positive correlation ($R^2 = 0.47$, $p < 0.05$) was recognized between the values of NDI(B8a − B7)/(B8a + B7) and NDF during the first pasture harvest survey (Figure 7d). Regarding NDF, there was also a large difference in the value range between the previous study [21] and our study. In our results, the NDF value range was from 50 to 70%, whereas in a previous study, the NDF value was 30–50%. Thus, the differences in the range of the NDF values may contribute to their relationships with the satellite-image-based indices.

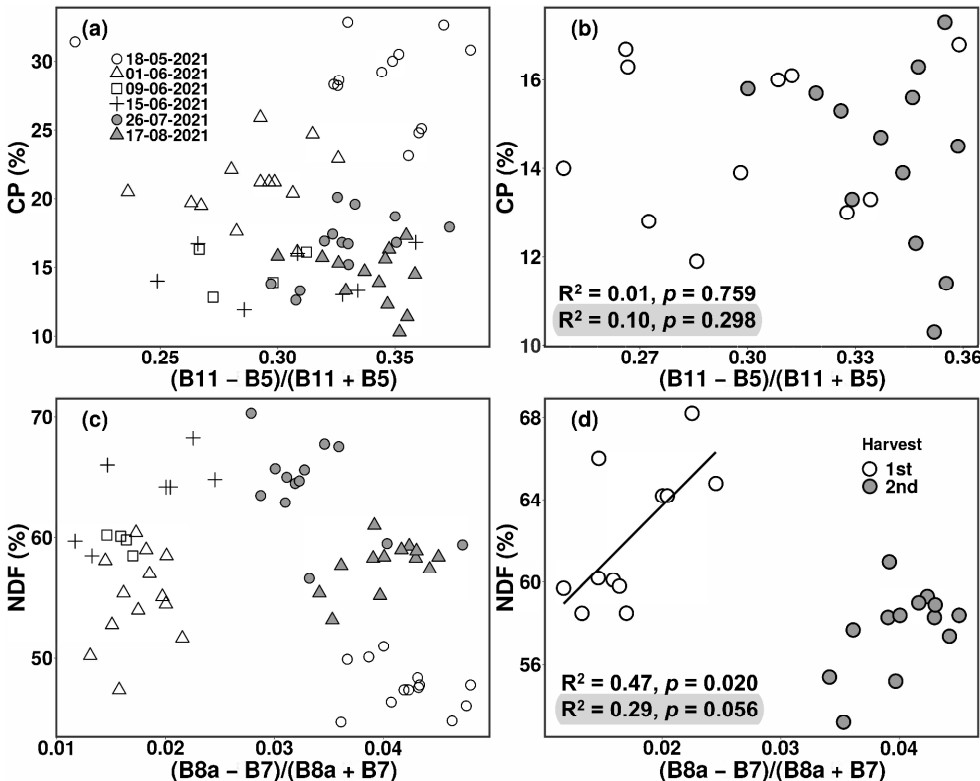

**Figure 7.** Relationships between NDI, CP, and NDF. (**a**): The relationship between $(B11 - B5)/(B11 + B5)$ and CP ($R^2 < 0.01$, $p = 0.466$), (**b**): the relationship between $(B11 - B5)/(B11 + B5)$ and CP at the harvest period, (**c**): the relationship between $(B8a - B7)/(B8a + B7)$ and NDF($R^2 = 0.09$, $p = 0.015$), (**d**): the relationship between $(B8a - B7)/(B8a + B7)$ and NDF at the harvest period. In (**a**,**c**), the shape of the plot indicates the survey date, and the gray points are the samples taken after the first pasture harvest. In (**b**,**d**), ○ indicates the first pasture harvest survey (9 and 15 June), and ● indicates the second harvest survey (17 August).

## 4. Conclusions

In this study, we aimed to predict the quantity and quality of hayfield pastures using Sentinel-2 satellite imagery in Kamishihoro, Hokkaido, Japan. Regarding the time course changes in pasture growth, the Enhanced Vegetation Index (EVI) predicted pasture growth better than the Normalized Difference Vegetation Index (NDVI). It is because the NDVI peaked in late spring and could not predict the late stage of pasture growth before the first harvest. We also compared the pasture biomass and quality (crude protein and neutral detergent fiber digestibility) data with the satellite images at specific timings for the first and second pasture harvest periods (early and late summer, respectively). At the first harvest timing, EVI could be used to predict the pasture biomass. At the first harvest, the index calculated based on Sentinel-2 bands 8a and 7 positively correlated to the neutral detergent fiber digestibility. However, these significant correlations were not found for the second harvest timing. During the second pasture harvest period, weed flowering and uneven pasture growth increased the heterogeneity of the pasture color within each paddock, making it challenging to select the representative spot for the ground-based pasture survey using a small (0.5 × 0.5 m) quadrat that we used. The heterogeneity of the paddock surface color is a reason for the weaker correlation between the satellite images to the ground survey data. To accurately predict the pasture yields when its growth is heterogeneous (the second harvest period), further studies should be performed with more ground sampling points.

**Author Contributions:** Y.U. developed the experimental plans. R.K. performed the experimental work with support from Y.U. Both R.K. and Y.U. contributed to the final version of the manuscript

and Y.U. supervised the whole project. All authors have read and agreed to the published version of the manuscript.

**Funding:** This work was funded by the Space Aerospace Science and Technology Promotion Commission (Fund ID 19196305), Ministry of Education, Cultural, Sports, Science and Technology, Japan.

**Institutional Review Board Statement:** Not applicable.

**Informed Consent Statement:** Not applicable.

**Data Availability Statement:** All of the data are available upon request to Y.U. (uchiday@agr.hokudai.ac.jp).

**Acknowledgments:** We thank the town office for Kamishihoro, Hokkaido and the agricultural cooperative for Kamishihoro, Hokkaido. From the two mentioned organizations, we particularly thank Hayashi and Kobayashi. We also thank Sawaguchi from Summit Agri-Business Corporation for the support of the field work.

**Conflicts of Interest:** The authors declare no conflict of interest.

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
