# Peer review of "Seasonal Changes in the Prediction Accuracy of Hayfield Productivity Using Sentinel-2 Remote-Sensing Data in Hokkaido, Japan"

_2813-3463, doi:10.3390/grasses2020006_

Round 1

Reviewer 1 Report

This study evaluated the use of remote sensing (NDVI and EVI) in assessment of hayfield yield and quality.  The experimental design and statistical analysis appear to be adequate for this study.  Overall, the manuscript is clear and well written and I feel that this work is an important contribution on this subject matter.  While significant relationships were found for yield and some quality parameters in the first harvest, correlations were not significant for the second harvest.  The technical reasons for this were adequately explained.  The flow of the manuscript is very good.

Author Response

Thank you very much for the review. Here, we upload the revised manuscript based on the other reviewers' comments (with the track changes). 

Reviewer 2 Report

The paper is very interesting, relevant and very well written. However, I believe that the paper is not suitable for publication due to a serious methodological error. Namely, the evaluation of grassland yield under field conditions was done by mowing the plant mass on an area of 0.5×0.5 m per experimental plot, "at one site per paddock". Sampling only 0.25 m2 of the area of plots with an area of 1.5-6.8 ha is not sufficient for accurate and precise evaluation of grassland yield.

Author Response

Thank you very much for the comment. We only took one quadrat sample per paddock for a few reasons. One was because we would like to mimic the grass assessment method applied by the local extension officers. Two was because the pasture was usually very uniform (especially from spring to summer), and we found mainly one grass species. Also, the paddocks were not disturbed by grazed animals (only used for silage production); thus, our experimental fields had no urine or dung patches. Three was because weeds (such as Docks) appeared as large patches in specific sections within a paddock. Thus, we believe it would be challenging to accurately quantify the paddock pasture yield even by increasing the quadrat samples per paddock, especially in the autumn season when the weeds start actively growing. 

We could not retake the samples, but we carefully explained the reasons above within the revised manuscript. Please see the materials and methods part (section 2.2) of the revised manuscript (attached). 

Thank you very much. 

Reviewer 3 Report

In this study, the author used satellite images to predict the biomass of forage. The main indicators adopted are NDVI and EVI. Sentinel-2 data can better predict the growth and quality of the region. The design of the article is reasonable, and the discussion and conclusion are sufficient. Spelling errors and grammatical problems should be carefully checked. It can only be published after further improvement.

Author Response

Thank you very much for the review. We have now carefully checked the spelling and grammatical errors using Grammarly Premium edition. The changed sections and words were shown as track changes. Please see the attached file. 

Round 2

Reviewer 2 Report

Regardless of the author's explanation, I continue to believe that a single sample from the mentioned area is insufficient for a accurate and precise assessment of grassland productivity.

Author Response

Thank you very much. We will discuss this matter with the editors. Thank you.